# School health professionals' understanding of culture: a scoping review protocol

Emmie Wahlström ,[1,2] Sara Landerdahl Stridsberg,[3] Camilla Larsson,[3] Jonas Stier[2]

[1]ChiP Research Group, Mälardalen University, Vasteras, Sweden
[2]School of Health, Care and Social Welfare, Mälardalen University, Vasteras, Sweden
[3]University Library, Mälardalen University, Vasteras, Sweden

**Correspondence to**
Dr Emmie Wahlström; emmie.wahlstrom@mdu.se

## ABSTRACT

**Introduction** Culture is highlighted in previous research as important in encounters where health professionals and children do not share a language or culture. In these encounters, culture is described as mainly related to the child, whereas the health professionals' understanding of their own culture as impacting the encounter tends to be left out. To clarify how culture is understood and conceptualised among professionals, it is of relevance to collate previous research on health professionals' understanding of culture. In the scoping review that this protocol describes, we aim to focus on the context of the school health services, being a context accessible to many children in their everyday life. The aim of the review will be to identify, describe and analyse previous research concerning school health professionals' (ie, school nurses, school social workers, school doctors and school psychologists) understanding of culture.

**Methods and analysis** This scoping review will be guided by the methodology described by Peters *et al* and Khalil *et al*. Searches will be conducted in Scopus, PubMed, Cinahl Plus, SocIndex, Sociological Abstracts, Social Services Abstracts, APA PsycInfo, APA PsycArticles, Web of Science and Applied Social Sciences Index & Abstracts (ASSIA). Any published scientific papers focusing on school health professionals' understanding of culture (conceptualised through a variety of related terms) and school health services conducted within the last 10 years (2013–2023) will be included. Two reviewers will independently screen all titles and abstracts for inclusion. Two reviewers will conduct the screening of full-text documents and the extraction of information. Qualitative content analysis as well as discourse analysis will be employed.

**Ethics and dissemination** Ethical approval is not required for this study. The findings will be disseminated through peer review publication as well as presentation at conferences and to relevant stakeholders.

## INTRODUCTION

Global trends suggest that more people have migrated than ever before. In 2020, about 281 million people were international migrants.[1] Most of these people were in the age of 15–64 years and about 15 per cent of migrants were below the age of 19 years. To provide high-quality healthcare services to migrants, the WHO states that culture is

---

## STRENGTHS AND LIMITATIONS OF THIS STUDY

⇒ The methodological description of scoping reviews by Peters *et al* and Khalil *et al* has been used to plan the review.
⇒ The searches will be conducted in 10 electronic databases to identify relevant studies published during the last 10 years (2013–2023).
⇒ The review will not include a formal assessment of study quality, as it is a scoping review.

---

an important factor in the service delivery of refugee- and migrant-sensitive health systems.[2] What does this statement entail? To what does culture refer to and how can it be conceptualised?

In political and scientific discourse alike, culture is alternatively conceptualised as an individual background variable, a cognitive scheme, a discursive construction or as a contextual characteristic. In encounters between migrant children and professionals working within health services, culture can be understood in all these conceptions. There is ample research recognising the importance of culture in such interaction,[3–5] yet in the health and welfare domain, the focus is primarily on culture as a background variable or cognitive scheme. Additionally, culture is sometimes highlighted in relation to challenges such as when the professionals and children *lack* a joint culture and/or common language.[5] By the same token, professionals tend to view culture mainly as a *characteristic of the child* in the encounter, whereas the professionals are conceived to be 'acultural',[4] or culture is considered as a background variable explaining various health outcomes and healthcare needs among the children.[6]

Moreover, the culture of a child and their family is also stressed as something to acknowledge in encounters. It is also stressed that culture of children and families should be considered when providing care and accounting for their needs.[3] Such 'cultural

considerations' encompass, for instance, ethnic traditions, cultural norms and values, or religious beliefs impacting on interaction in general and the provision of healthcare in particular. 'Cultural considerations' also include aspects related to limited resources available in low-income or rural areas, as well as language and communication 'barriers'. In relation to the 'cultural considerations' of children, findings in the same review of healthcare for paediatric patients show that the education of healthcare professionals tends to be ethnocentric, Eurocentric and without cultural self-reflection, that is, what Stier describes as discursive blind spots or biases.[7] Thus, rather than counteracting 'problems' or misunderstandings or merely focusing on the culture of children and parents, such biases in education or among professionals, for instance, in school healthcare services, may fuel constructions of other needs as 'different' leaving healthcare professionals without competence to encounter children with needs, values and cultures outside 'the norm'.[8]

Against this background, there is a need for research addressing culture as a discourse, social construction and interactional accomplishment as well as a need to collate research on the culture(s) within the healthcare services targeting children's health and well-being beyond the paediatric context. One such healthcare service devoted to promoting health and preventing disease among children is the school health services (SHS). In an international context, SHS is multiprofessional, although the most common profession is school nurses.[9] Among other professions, it includes doctors, psychologists and social workers. In Sweden, SHS consists mainly of school nurses, school doctors/physicians, school social workers, school psychologists and special education teachers. Out of these, school nurses are the largest number of professionals working solely within SHS (2910 persons in year 2021) followed by school social workers (2830 persons in year 2021).[10] These two professional groups often have an office at the school and are available for spontaneous drop-in meetings where the children can discuss issues relating to their health and/or well-being. School doctors and school psychologists mainly serve as consultants, often to several schools within a limited geographical area (municipality or region). Although the work content of each profession contains aspects unique to the Nordic and Swedish context, similarities have been found with professionals' work internationally.[9 11 12] School nurses and school doctors worldwide encounter children to assess and screen the children's health and development, vaccinate children against diseases and provide health education.[9] Similarly, school social workers worldwide encounter children facing similar problems, for example, physical and psychological health issues, poverty, issues concerning attendance, dropout and motivation as well as decaying neighbourhoods and racism.[12] Also, school psychologists are available in many countries[13] and work with similar activities.[14] By being available in the school setting, these professionals can assist children with issues related to health and development in the children's everyday life, making it interesting to scientifically investigate these professionals' understanding of culture when interacting with children.

Previous studies have been conducted studying preschool teachers', teachers', school nurses', school social workers' and school psychologists' encounters with children, and have highlighted culture as relevant, especially in encounters with children who have migrated[15–20] and in relation to race or ethnicity.[21] Yet there is a lack of reviews collating research regarding school health professionals' (ie, school nurses, school social workers, school doctors and school psychologists) understanding of culture. A preliminary search for existing scoping reviews within this area of research has been conducted by the research group (SLS and CL) in the Cochrane Library, Epistemonikos, Internationa HTA database, Open Science Framework, Social Care Online and PROSPERO. The searches confirmed that no scoping reviews are or have been conducted previously on this topic.

## Objective

This review aims to identify, describe and analyse existing research on school health professionals' (ie, school nurses, school social workers, school doctors and school psychologists) understanding of culture.

## Review questions

► What research has been conducted on school nurses', school social workers', school doctors' and school psychologists' understanding of culture?
► What research has been conducted on school nurses', school social workers', school doctors' and school psychologists' cultural self-understanding?
► What research has been conducted on intercultural interaction and communication in the context of school health professionals' encounters with students and their families?
► In what geographical contexts (ie, countries) have research concerning school nurses', school social workers', school doctors' and school psychologists' understanding of culture been conducted?
► How has culture been conceptualised in research concerning school nurses', school social workers', school doctors' and school psychologists' understanding of culture?

## METHODS
### Patient and public involvement statement
The scoping review protocol has been developed without any patient or public involvement. This scoping review is not planned to include patient or public involvement.

This scoping review protocol follows the structure described by Peters *et al* and the Joanna Briggs Institue (JBI) manual.[22 23] The methodology described by Peters *et al* and Khalil *et al* has guided the design of the scoping review and how it will be conducted.[22 24] The scoping review protocol has been planned and designed in

collaboration with research librarians (CL and SLS), who will also conduct the searches for the scoping review.

## Eligibility criteria (participants, concepts, context and types of evidence source)

Eligibility criteria for this scoping review are based on the format suggested by Peters *et al* and the JBI manual focusing on defining the participants, the concept and the context.[22 23] In this scoping review, research concerning the four *types of participants* included in the aim will be included, for example, school nurses, school social workers, school doctors and school psychologists. The profession of school nurse, school doctor or physician and school psychologist has the same 'title' of the profession regardless of geographical context, although the content of what they do might differ somewhat.[9 11 13 14] However, for the profession of school social worker, the 'title' of the profession varies between geographical contexts.[12] To account for these variations, the variation of 'titles' described in Beck and Hämäläinen will be included in the searches. In addition, the descriptions of the profession in the articles generated from the searches will be checked in relation to descriptions of the profession of 'school social worker'. As counselling is part of the school social workers' practice search, terms related to school counselling will be included, although the profession of school counsellor is not included in the scoping review.

The *concept* in focus in the scoping review is culture, more specifically professional's understanding of culture. Therefore, eligible articles for this review are those that mention professional's understanding of culture in the title, abstract, keywords or findings. However, no particular definition of culture will guide the inclusion of articles as one of the research questions concerns how culture has been conceptualised in the studies. Eligible articles are articles that are located in the *context* of SHS. As school systems vary between countries, eligible articles will be articles that include school health professionals working at schools or regularly visiting schools where children and adolescents aged 6–19 years attend. This implies that articles including school health professionals working in schools that enrol students at university level (or equivalent level) are not included in the review. The *types of evidence sources* used for this scoping review are published scientific papers, including both reviews and primary studies as well as descriptive articles presenting theoretical frameworks, models or concepts. To avoid duplicate data, primary studies will be excluded if their results have been included and reported in the results of an included review. An additional limitation of the searches is that the publication date should be within the last 10 years (2013–2023) to reflect the contemporary research field.

## Search strategy

As studies of school health professionals might be published in journals with varying scopes, from medicine to psychology and social work, a wide range of databases will be included. Searches will be conducted in Scopus (https://www.scopus.com/), PubMed (https://pubmed.ncbi.nlm.nih.gov/), Cinahl Plus (EBSCOhost), SocIndex (EBSCOhost), Sociological Abstracts (ProQuest), Social Services Abstracts (ProQuest), APA PsycInfo (EBSCOhost), APA PsycArticles (EBSCOhost), Web of Science (https://www.webofscience.com) and Applied Social Sciences Index & Abstracts (ASSIA) (ProQuest). The search terms (see table 1) will be combined using the Boolean operators AND and OR, and phrase searching and truncation will be applied where appropriate. The search strategy will include a combination of free text terms and, where applicable, controlled vocabulary which relates to the eligibility criteria of the review (participants, concept and context). The full initial search strategy is provided in table 2.

| Table 1 | Search terms used based on the eligibility criteria | |
|---|---|---|
| **Eligibility criteria** | **Search theme** | **Search terms used** |
| Participants | School nurses | "school nurs*" |
| | School doctors | "school doctor*", "school physician*" |
| | School psychologists | "school psych*" |
| | School social workers | "school social work*", "school welfare officer*", "education social worker", "education welfare officer*", "attendance counselor", "school counsel*", "guidance counselor*" |
| Concept | Understanding of culture | "cultural* competen*", "cultural* sensitiv*", transcultural, "cross cultural", "cultural* congruent*", "cultural care", multicultur*, "cultural divers*", intercultural, "cultural awareness", "cultural nursing", "cultural communication", "indigenous knowledge", "cultural skill*", "cultural understanding", "cultural interaction*", "cultural knowledge", "cultural proficienc*", "cultural dynamic*", "cultural safety", "cultural meeting", "cultural bias", "cultural self*", "cultural identification" |
| Context | School health services | "school health", "school based health", "school mental health service*", "school based mental health service*" |

\* indicates how truncation have been used for included search terms.

In accordance with Peters *et al*,[22] the searches may be modified and expanded during the process to account for any new potentially relevant terms, concepts or contexts based on the results of searches or the process of evidence selection. Conducted searches will also be accompanied by manual searches of relevant references cited in the included studies, in the Web of Science Core Collection, Scopus and Google Scholar.

### Study/source of evidence selection

To assist in the evidence selection process and as a tool for deduplication, the Covidence software will be used. The selection of sources will begin by screening the title and abstract of each publication in the search results. Initial screening will be conducted by EW and JS for titles and abstracts against the inclusion criteria. The inclusion of articles will be guided by the population, concept and context as well as types of evidence described above in the eligibility criteria. This process will also assist in the clarification of inclusion and exclusion criteria if needed. Agreement in the selection process will be determined through discussions between authors regarding the relevance of selected publications to the eligibility criteria until negotiated consensus is reached. Thereafter, the included full-text publications will be examined by EW and JS. Calibration of inclusion between reviewers will be conducted by selecting 5 per cent of the articles for screening by both reviewers and control for agreement in decisions at both the initial and the full-text stages. An agreement of 90 per cent is strived for, and if this is not reached, disagreements will be discussed and inclusion criteria revised accordingly.[25]

Exclusion of publications will be shown using a flow diagram or chart such as the adapted chart in the PRISMA-ScR. Studies will be excluded if full-text documents cannot be retrieved. In addition, articles will also be excluded if they focus on culture in intraprofessional or interprofessional encounters, describe culture in terms of art (dance, painting, music, etc) or are published as editorials, debates, calls to action, commentaries or similar texts. Excluded publications and reasons for exclusion will also be reported in an appendix for all publications examined in full text.

### Data extraction

Data from the included full-text publications will be processed using the Covidence software and exported to a Microsoft Excel spreadsheet containing the following headings: author(s), year of publication, where the source was published or conducted (country of origin), aims/purpose, population and sample size, methodology/methods, outcomes and details of these (if applicable), conceptualization of culture as well as key findings that relate to the scoping review questions. If needed during the charting process, the data extraction spreadsheet might be updated to include additional headings to better capture content relevant to the scoping review. Extraction will be conducted by EW and JS. Calibration of

| **Table 2** | Initial search strategy |
| --- | --- |
| **Blocks** | **Search string/search terms** |
| 1. | TITLE-ABS-KEY ("school health" OR "school based health" OR "school nurs*" OR "school mental health service*" OR "school based mental health service*" OR "school social work*" OR "school welfare officer*" OR "education social worker" OR "education welfare officer*" OR "attendance counselor" OR "school counsel*" OR "guidance counselor*" OR "school doctor" OR "school physician" OR "school psych*") |
| 2. | TITLE-ABS-KEY ("cultural* competen*" OR "cultural* sensitiv*" OR transcultural OR "cross cultural" OR "cultural* congruent*" OR "cultural care" OR multicultur* OR "cultural divers*" OR intercultural OR "cultural awareness" OR "cultural nursing" OR "cultural communication" OR "indigenous knowledge" OR "cultural skill*" OR "cultural understanding" OR "cultural interaction*" OR "cultural knowledge" OR "cultural proficienc*" OR "cultural dynamic*" OR "cultural safety" OR "cultural meeting" OR "cultural bias" OR "cultural self*", "cultural identification") |
| Combination | 1 AND 2 |
| Limitations | 2013–2023 |

\* indicates how truncation have been used for included search terms.

extraction will be conducted by 5–10 articles being independently processed by both reviewers. Discrepancies in extraction will be discussed until negotiated consensus is reached and the extraction form will be adjusted accordingly if needed.

### Data analysis and presentation

Data analysis will consist of summarising the extracted data related to methodology, professions and geographical contexts to answer related research questions. In addition, extracted data related to the conceptualisation of culture and key findings will be analysed using qualitative content analysis and discourse analysis. The results will be described in relation to the research questions and as appropriate in relation to each analysis methodology. The results of the discourse analysis of the included publications will be presented in a separate publication.

### ETHICS AND DISSEMINATION

This review does not require ethical approval. The dissemination strategy includes peer review publication and presentation at conferences and to relevant stakeholders.

**Contributors** EW and JS conceptualised the idea and study design. SLS and CL contributed to discussions on study design, conducted searches of existing

literature reviews and conducted initial test searches to clarify search terms and search strategies. EW drafted the first version of the protocol. JS, SLS and CL critically commented and/or revised the text. Furthermore, all authors have given final approval of the version to be published and agree to be accountable for all aspects of the work.

**Funding** The authors have not declared a specific grant for this research from any funding agency in the public, commercial or not-for-profit sectors.

**Competing interests** None declared.

**Patient and public involvement** Patients and/or the public were not involved in the design, or conduct, or reporting, or dissemination plans of this research.

**Patient consent for publication** Not applicable.

**Provenance and peer review** Not commissioned; externally peer reviewed.

**ORCID iD**
Emmie Wahlström http://orcid.org/0000-0002-7828-6999

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
