## [Reviewer comments · BMJ Open]

ARTICLE DETAILS

TITLE (PROVISIONAL)	School health professionals' understanding of culture: a scoping review protocol
AUTHORS	Wahlström, Emmie; Landerdahl Stridsberg, Sara; Larsson, Camilla; Stier, Jonas

VERSION 1 – REVIEW

REVIEWER	Baltag , Valentina World Health Organisation, Department of Maternal, Newborn, Child & Adolescent Health & Ageing
REVIEW RETURNED	10-Oct-2023

GENERAL COMMENTS	School nurses' and school social workers' understanding of culture: a scoping review protocol.	
	Review by Valentina Baltag, MD, MSc, PhD, World Health Organization	
	Aspect of the paper	Comment and suggestions
	The importance of the research question	This study protocol addresses a very timely and practical research question, namely the importance of understanding how culture influences encounters with children that come from different cultural backgrounds. In the context of raising migration answering this question has practical value.
	Methods – partial link between research question and methods	In my view the most important – and neglected - aspect of understanding how culture influences encounters with children that come from different cultural backgrounds is the aspect of health care professionals' own culture and self-awareness about it. As you rightly point, understanding culture by health care professionals is described as mainly related to the child culture whereas the health professionals' understanding of their own culture as impacting the encounter tends to be left out. This is mentioned in the introduction but is not followed up in methods and research questions. The methods and research questions are designed to describe understanding culture in general, and risk not to capture the aspect of self-awareness of own culture and its influences on the encounter. I suggest

		refining methods to ensure this particular aspect is captured.
	Methods – partial link between research question and methods	The study aims inter alia to highlight gaps in knowledge related to school nurses' and school social workers' understanding of culture. What is the standard for knowledge? Are requisite knowledge known? The methods section does not describe any tools or frameworks by which this question can be answered, suggest to strengthen methods so it becomes an answerable question
	Concept – school health services	More nuanced definition of school health services is required. School health services can be school based or school linked. This can mean different things in different countries, e.g. being school based full time, or school-based part time, or visiting schools on a pre-defined schedule. Please clarify which situations will be included
	Concept - participants	School nurses and social workers are included based on argument that these are the most common cadres in Sweden. However, the research is global in scope therefore other cadres should be considered. School psychologists as a very common cadre in school health services. The proposed terms are unlikely to capture this cadre. The other commonly reported cadre is a school doctor. If the intention is to include both school-based and school linked services (see my comment above), then I suggest school doctors be included as well (since many school-linked services have school doctors as common cadre).
	Methods – reviewers	It is suggested that only one reviewer will conduct data extraction, and the other one will validate. This is vague. A calibration exercise both at the stage of titles and abstract review as well as full text needs to be explained (e.g. in terms of sample size for validation, level of consensus to be reached). Also, if the yield of papers will be not too high (which I expect might be the case), better to consider 2 people extracting data and reconciling differences, to uphold a gold standard of data extraction by at least two reviewers to reduce the chance of errors and bias.
	Methods – data extraction	It is proposed that data will be extracted for “conceptualization of culture”. This level of framing the element for data extraction is too broad even for the start. I suggest using an a priori framework for key categories in conceptualization of culture, and then refine it based on your findings, through an iterative process of finalizing your data extraction form. In relation to my previous comment about the specific aspect of lack of awareness of own culture, I suggest to

	explicitly include items in data extraction form related to this.
--	---

VERSION 1 – AUTHOR RESPONSE

Reviewer 1 comments	Response by authors
This study protocol addresses a very timely and practical research question, namely the importance of understanding how culture influences encounters with children that come from different cultural backgrounds. In the context of raising migration answering this question has practical value.	Thank you.
In my view the most important – and neglected - aspect of understanding how culture influences encounters with children that come from different cultural backgrounds is the aspect of health care professionals' own culture and self-awareness about it. As you rightly point, understanding culture by health care professionals is described as mainly related to the child culture whereas the health professionals' understanding of their own culture as impacting the encounter tends to be left out. This is mentioned in the introduction but is not followed up in methods and research questions. The methods and research questions are designed to describe understanding culture in general, and risk not to capture the aspect of self-awareness of own culture and its influences on the encounter. I suggest refining methods to ensure this particular aspect is captured.	In the previous version, this was vaguely described. Therefore we have made this more explicit by talking about cultural self-understanding in the research question and adding related search terms.
The study aims inter alia to highlight gaps in knowledge related to school nurses' and school social workers' understanding of culture. What is the	Thank you for highlighting this. We have removed this part of the aim and the related research question to make room for adding a research question specifically directed towards

standard for knowledge? Are requisite knowledge known? The methods section does not describe any tools or frameworks by which this question can be answered, suggest to strengthen methods so it becomes an answerable question	self-understanding as well as the inclusion of school doctors and school psychologists.
More nuanced definition of school health services is required. School health services can be school based or school linked. This can mean different things in different countries, e.g. being school based full time, or school-based part time, or visiting schools on a pre-defined schedule. Please clarify which situations will be included	Our intention is to include both school-based and school-linked and we have revised the text to include both these concepts citing your study, see page 5, line 155.
School nurses and social workers are included based on argument that these are the most common cadres in Sweden. However, the research is global in scope therefore other cadres should be considered. School psychologists as a very common cadre in school health services. The proposed terms are unlikely to capture this cadre. The other commonly reported cadre is a school doctor. If the intention is to include both school-based and school linked services (see my comment above), then I suggest school doctors be included as well (since many school-linked services have school doctors as common cadre).	We agree that this would be a relevant addition. The review aim, research questions and design have been revised to include school psychologists and school doctors.
It is suggested that only one reviewer will conduct data extraction, and the other one will validate. This is vague. A calibration exercise both at the stage of titles and abstract review as well as full text needs to be explained (e.g. in terms of sample size for validation, level of consensus to be reached). Also, if the yield of papers will be not too high (which I expect might be the case), better to consider 2 people extracting data and reconciling differences, to uphold a gold standard of data extraction by at least two reviewers to reduce the chance of errors and bias.	Thank you for asking us to clarify this. The sections describing study/source of evidence selection and data extraction has been revised according to this and similar comments provided.

It is proposed that data will be extracted for “conceptualization of culture”. This level of framing the element for data extraction is too broad even for the start. I suggest using an a priori framework for key categories in conceptualization of culture, and then refine it based on your findings, through an iterative process of finalizing your data extraction form. In relation to my previous comment about the specific aspect of lack of awareness of own culture, I suggest to explicitly include items in data extraction form related to this.	Drawing upon a postcolonial perspective and constructionist epistemology, we view culture more as an emergent rather than an a priori concept. Hence, the conceptualization used in generated articles will guide us in how the extraction will be made.
Abstract: how this (validation by other reviewer) will be done is not currently explained in the main text, please explain	This sentence has been revised as two reviewers will instead conduct the screening and extraction processes. Related text in subsequent sections has also been clarified according to provided comments.
Introduction: One key argument why this study is important, is that migration is on the rise, and many more children are attending schools in countries outside their home culture. Suggest to provide some key statistics illustrating this	Some key statistics have been included in the manuscript.
Introduction line 31-33: “Cultural needs” are explained but not culture-mediated expectations from health-care. Expectations are quite different from needs therefore I suggest to address them separately. E.g. your point about low-income and rural areas might related to diverse needs, but also different expectations	The paragraph containing this sentence has been revised to clarify content.
Review questions: In the Background section your most interesting rationale for this study is the fact that health-care professionals are less self-aware of how their own culture influences the encounter (while being generally more aware of the students' culture). I do not see this aspect reflected in your questions.	A new research question has been added to address this specific aspect.
Eligibility criteria p 6, lines 23-25: those are two different things. An article can mention culture in the text without aiming to study culture. Please clarify if you will include both type, or just the latter	Yes, thank you for pointing this out. We have clarified that we will only include articles that contain culture in the title, abstract, key words, or findings. This would allow for articles who have included findings related to culture without aiming to study culture to be included. Articles that merely mention culture as part of the discussion of findings or suggestion of further research will not be included.
P 6. Line 30 working in schools - this can mean different things in different countries, e.g. being school based full time, or school-based part	Revised according to similar comment above.

time, or visiting schools on a pre-defined schedule. Please clarify which situations will be included	
P. 6, line 34-35 including reviews and primary studies: does this mean that descriptive articles will be excluded? Given the concept in focus, many articles are likely to be descriptive articles without including a research study	Thank you for this comment. We plan to include publications that report a conducted study (regardless of the design or type of data used for this study) as well as articles presenting a theoretical framework, concept or model. We will not include editorials, commentaries, debates, calls to action, or similar types of texts.
Table 1: Participants: suggest to include school psychologists as a very common cadre in school health services. The proposed terms are unlikely to capture this cadre. The other commonly reported cadre is a school doctor. If the intention is to include both school-based and school linked services, then I suggest school doctors be included as well (since many school-linked services have school doctors as common cadre)	Revised according to similar comment above.
Table 1: school social workers: suggest to add "health educator"	We have not seen that school social workers have been referred to as health educators in previous studies. As the terminology for this group varies between geographical contexts, we have used the terminology from Beck & Hämäläinen (2022) which describe the most common "titles" which adheres to a similar description of the services provided by a school social worker globally. Health educator seems to refer to another type of professional working in various settings beyond the school health services.
Table 1 concept: suggest to add cultural bias, cultural self-awareness, self-reflection	Three new search terms have been added: cultural bias, cultural identification and cultural self*. The latter will search hits including all variations starting with cultural self, i.e. cultural self-awareness, self-reflection, self-understanding, etc.
Table 1 context: suggest to add school health center, school health clinic, school health service, school health care	The search term "school health" will capture all variations of school health services starting with "school health" such as school health center / care / clinic / service / etc.
Table 2 search string/search terms: please consider including other terms as listed above	Revised in relation to comments provided above.
Study/source of evidence selection, p 7, lines 12-17: more details need to be provided how the calibration exercise will be conducted. E.g. which level of agreement do you want to reach? 90% or more is recommended	The text has been revised to clarify planned calibration exercise.
P. 7 line 20: is this the only a priori exclusion criteria? If not please list the other	Two additional exclusion criteria have been added for clarity.
Data extraction p7, line 32: this is too broad even for the start. Suggest to use an a priori	See comment above.

framework for key categories in conceptualization of culture, and then refine it based on your findings, through an iterative process of finalizing your data extraction form. In relation to my previous comment about the specific aspect of lack of awareness of own culture, suggest to explicitly include items in data extraction form related to this	
P. 7, lines 36-37: please explain in more details (i) what validation means - parallel data extraction? random check? what level of consensus do you aim to reach?	The text has been revised to clarify planned calibration exercise.
Conclusion, p. 7, lines 52-53: nothing from the described above will answer this question. The protocol needs to be strengthened so that the research can answer this question. Inter alia, gaps analysis requires a standard against which the reality is compared. This would require that the protocol is explicit about what the standards are.	The conclusions section has been removed as requested by the associated editor.

VERSION 2 – REVIEW

REVIEWER	Baltag , Valentina World Health Organisation, Department of Maternal, Newborn, Child & Adolescent Health & Ageing
REVIEW RETURNED	01-Nov-2023
GENERAL COMMENTS	Most of my suggestions for improvement were adequately addressed, with one exception regarding the use of a priori framework to categorize elements of culture. However the explanation provided is sufficient to accept this alternative course of action. I am looking forward to see the results of this interesting study.